# Impact of Dropping on Postharvest Physiology of Tomato Fruits Harvested at Green and Red Ripeness Stages

**DOI:** 10.3390/biom14081012

**Published:** 2024-08-15

**Authors:** Chy Sophea, Nasratullah Habibi, Naoki Terada, Atsushi Sanada, Kaihei Koshio

**Affiliations:** 1Graduate School of Agriculture, Tokyo University of Agriculture, Tokyo 156-8502, Japan; sophea8250@gmail.com (C.S.); nt204361@nodai.ac.jp (N.T.); a3sanada@nodai.ac.jp (A.S.); koshio@nodai.ac.jp (K.K.); 2Faculty of Agriculture, Balkh University, Balkh 1701, Afghanistan

**Keywords:** dropping, postharvest physiology, ripeness stage, tomato

## Abstract

Dropping during transportation is a critical issue for tomato fruits, as it triggers ethylene production and affects quality parameters, leading to lower quality and a reduced storage life. Thus, this study was conducted to assess the physiological alterations in tomato fruits subjected to dropping. This study involved tomatoes harvested at green and red stages, subjected to the following five dropping treatments: 0 cm, 10 cm, 30 cm, 50 cm, and 100 cm. The results revealed that dropping from 100 cm induced the highest ethylene production, particularly in green fruits, where production began within one hour and peaked within 48 h. Red fruits exhibited a dose-dependent response to mechanical stress, with a notable decrease in ethylene production starting from the second week post-dropping, suggesting a regulatory mechanism. CO_2_ production peaked at 350.1 µL g^−1^ h^−1^ in green fruits and 338.2 µL g^−1^ h^−1^ in red fruits one day after dropping from 100 cm. Dropping also significantly influenced fruit color, firmness, electrolyte leakage, and vitamin C content. Principal component analysis (PCA) revealed distinct changes in metabolite profiles, with methionine and ACC (1-aminocyclopropane-1-carboxylate), key ethylene precursors, increasing in response to dropping, particularly in red fruits. These findings underscore the critical role of mechanical stress in modulating fruit physiology, with implications for post-harvest handling practices aimed at enhancing fruit quality and shelf life.

## 1. Introduction

Tomatoes (*Solanum lycopersicum* L.) are globally significant due to their rich nutritional profile, including vitamin A, vitamin C, and antioxidants [1,2,3]. Despite their nutritional value, local production in developing countries often falls short of market demands, leading to substantial imports from neighboring countries [4,5]. The postharvest journey of tomatoes in hot–wet tropical climates is particularly challenging, resulting in significant losses during transportation due to mechanical stresses such as dropping and throwing [6,7,8]. Tomatoes are especially sensitive and perishable when subjected to vibration or dropping during transit. Research indicates that mechanical injuries during transportation [9] not only cause physical damage but also affect the internal quality [10,11,12] of tomatoes, particularly their flavor [13]. Studies have shown that mechanical damage leads to reductions in acid content and vitamin C levels, as well as altered ethylene production [14,15,16]. The negative impacts of mechanical stress on tomatoes extend beyond immediate physical damage [17,18,19]. There are cascading effects, such as increased ethylene production leading to accelerated ripening and a reduced shelf life. Weight loss due to moisture loss, changes in firmness [20,21], enzymatic activities, and alterations in fruit color [22] and total soluble solids are also notable outcomes of mechanical stress during postharvest handling.

Bruising, scarring, and other mechanical injuries during harvesting, sorting, packing, and transportation cumulatively degrade tomato quality, impacting its marketability and economic value for farmers and marketers alike [23]. Transportation poses a critical challenge in maintaining tomato quality [24], especially in developing countries with poor road infrastructure and limited access to refrigerated transport options [25]. Mechanical damage during transit due to vibrations and impacts further exacerbates postharvest losses. Strategies such as proper packaging, careful handling, and suitable transportation modes are essential to mitigate these losses and ensure that fresh produce reaches consumers in optimal condition [26]. Understanding the physiological aspects of tomato ripening, especially the role of ethylene, is crucial in managing postharvest quality [27]. Ethylene production is influenced by various developmental and external factors, including ripening, senescence, and mechanical injuries [28,29].

This hormone’s biosynthesis pathway in fruit tissues involves complex enzymatic reactions, highlighting the intricate processes involved in postharvest physiology [30]. Respiration, another essential metabolic activity, also plays a role in ethylene production and fruit ripening processes, especially in the case of climacteric fruits like tomatoes [31,32]. In tomato fruits, the quality strongly depends on the metabolites [33].

The physical, physiological, and quality attributes of tomato fruits are significantly impacted by dropping from various heights during transportation. While some studies have explored the effects of vibration on fruit quality during transport and others have examined how bruising affects the postharvest quality of tomatoes, there remains a gap in understanding the specific impacts of dropping from different heights. This lack of knowledge is particularly concerning, given the common occurrence of such mechanical stress during the handling and transport of tomatoes. Tomato fruits can be harvested at different stages of ripeness, typically categorized as green and red stages. The physiological responses and quality attributes of tomatoes at these stages may differ when subjected to mechanical stress. However, research specifically focusing on how dropping from different heights affects tomatoes harvested at these distinct stages is sparse.

Therefore, this study was conducted to fill this knowledge gap by assessing the effects of mechanical stress from dropping on the postharvest physiology and metabolites of tomatoes harvested at both the green and red stages. By understanding how different heights of dropping impact these factors, the study aims to provide insights that can help improve handling and transportation practices, ultimately enhancing the postharvest quality and marketability of tomatoes.

## 2. Materials and Methods

### 2.1. Experimental Setup

The study was conducted at the Tropical Horticultural Science Laboratory of Tokyo University of Agriculture from 2022 to 2023. For the experiments, one hundred uniform tomato fruits were selected based on their initial ethylene production levels at Green Farm Lapin, Namegawa-cho, Saitama Prefecture, and transported to the laboratory immediately. The fruits were then allocated into five treatment groups: (1) 0 cm (control): samples that were not dropped; (2) 10 cm: samples dropped from a height of 10 cm; (3) 30 cm: samples dropped from a height of 30 cm; (4) 50 cm: samples dropped from a height of 50 cm; and (5) 100 cm: samples dropped from a height of 100 cm. Following the treatment, all samples were stored at 25 °C in an incubator (SANYO, MIR-253, Osaka, Japan).

### 2.2. Application of Mechanical Damage

Tomato fruits were dropped once from heights of 10 cm, 30 cm, 50 cm, and 100 cm, with a no-dropping treatment serving as the control. No visible damage was observed on the surfaces of the fruits following treatment. The impacts were distributed evenly across the equatorial surface of the fruits, with different areas of the equatorial surface contacting the concrete floor with each impact. One day after the dropping treatment, ten tomato fruits from each group were placed in 550 mL glass jars and incubated at 25 °C. Respiration and ethylene production rates were measured after 1, 4, and 7 days. Each jar was sealed tightly for at least 1 h prior to measurements, and the respiration and ethylene production rates were determined based on the increases in carbon dioxide and ethylene concentrations, respectively.

### 2.3. Ethylene Production Measurement

For the measurement of ethylene, all the tomato fruits harvested at green and red stages of the same size and color were weighed and initial ethylene production was measured before the final selection; the fruits were then sorted based on the initial ethylene production. The tomato fruits were enclosed in 550 mL jars for one hour under a black sheet of cloth to prevent the light affecting them; one ml of gas was extracted using a plastic syringe. The gaseous sample was injected to the GC-FID (gas chromatography–flame ionization detector) (GC-14B), and the data were expressed as nL g^−1^ h^−1^ for further analysis [34,35].

### 2.4. Quantification of Ethylene Production Capacity of ACC

Disks of fruit tissues, measuring 5 mm in diameter, were taken out with a cork borer; four disks were taken from slices of three fruits per treatment and placed in ice cube trays. Then, 5 mL drops of water (control), 1 mM, and 10 mM ACC were applied to each of the disks. The disks were saturated with ACC solution and soaked for 1 h. Finally, the disks were transferred into bottles, then incubated (25 °C) for 1 h. Later, 1 mL of gas from the headspace was withdrawn for analysis of the ethylene contents. Ethylene was detected using gas chromatography, specifically GC-FID (gas chromatography–flame ionization detector), (Agilent Technologies Inc., Santa Clara, CA, USA) and data were expressed as nL g^−1^ h^−1^.

### 2.5. CO_2_ Release Measurement

For the measurement of carbon dioxide, all the tomato fruits harvested at green and red stages of almost the same size and color were weighed, and the initial ethylene production was measured prior to the final selection and sorting of the fruits. After the fruits were dropped, they were enclosed in 550 mL jars for one hour under a back sheet of cloth to prevent the light affecting them; one ml of gas was extracted using a plastic syringe. The gaseous sample was injected into the GC-TCD (gas chromatography–thermal conductivity detector) (Agilent Technologies Inc., CA, USA), and the data were expressed in µL × g^−1^ × h^−1^ for the further analysis [34].

### 2.6. Color Change

The degree of ripening was generally estimated using color charts and colorimeters, which express color in numerical terms using the a* value (from green to red) [36,37,38]. In this experiment, pigmentation was measured using Colorimeter NR-3000. Data were recorded every day.

### 2.7. Fruit Firmness Measurements

Fruit firmness was checked using a penetrometer (Instron 3342, Illinois Tool Works Inc., Hopkinton, MA, USA). The measurements were taken every day until the end of the experiment (14 days). The data were expressed as N.

### 2.8. Brix Measurement

For the measurement of TSS (Brix), all the tomato fruits harvested at green and red stages of almost the same size and color were weighed, and the initial ethylene production was measured before the final selection and sorting of the fruits. After that, 10 replications of fruits were dropped from 0, 10, 30, 50, and 100 cm, and the tomato fruit was sliced (with the jelly and seeds removed) and crushed to make juice at 4 °C. After filtration, using two layers of mesh cloth, and homogenization, a refractometer was used for the measurement of the Brix levels (%).

### 2.9. Vitamin C

For the measurement of vitamin C or ascorbic acid, all the tomato fruit harvested at green and red stages of almost the same size and color were weighed, and initial ethylene production was measured before the final selection and sorting of the fruits. An amount of 2 g was weighed out of the tomato fruits. The jelly and seeds were removed and the fruits were crushed in meta phosphoric acid (5%) on ice to make juice at 4 °C. After filtration, using two layers of mesh cloth, and homogenization, 2 mL of mixed juice was taken and put into small tubes. The samples were centrifuged for 5 min at 120,000 rpm and 25 °C; the reflectometer RQ Flex 10 was used for the measurement of ascorbic acid, and it was expressed as mg/100 g of FW (fresh weight).

### 2.10. Electrolyte Leakage Measurement

Tomato fruits were sliced, and 3 pericarp disks (0.5 cm in diameter) were taken from the equatorial region of each fruit. The jelly and seeds were removed, and the fruits were washed with deionized water. The tomato disks were put into distilled water for 1 h and their electrical conductivity (EC) was measured. After that, the samples were placed into heating block EB-603 for 1 h of boiling. The samples were put into the Vortex for mixing samples and their EC was evaluated. Finally, the data were recoded as µS/cm and transformed into percentages [39,40].

### 2.11. Metabolomic Analysis

Metabolomic analysis was performed using a GC/MS (Shimadzu) following a modified method described by Jeong et al. [41]. To investigate the effects of dropping on tomato fruit metabolites, samples were collected one week post-dropping for comparison between dropped and non-dropped fruits. Whole tomato fruits were homogenized using a mortar and pre-cooled with liquid nitrogen, and 100 mg of the homogenized tomato powder was used to prepare the samples. Each sample was combined with one zirconia bead and 250 µL of methanol. The samples were mixed thoroughly in a mixer mill MM400 for 2 min at 27 Hz. Subsequently, 250 µL of chloroform was added, and the samples were incubated in a thermomixer for 3 min at 37 °C and 1200 rpm. Standard solution (50 µL) and ultra-pure water (125 µL) were then added to each sample, which was centrifuged at 1500× *g* for 10 min at 25 °C. A careful extraction of 80 µL of the supernatant was transferred to a 1.5 mL Eppendorf tube and evaporated for 2 h. The samples were then placed in a freeze dryer and stored overnight. Following lyophilization, 40 µL of methoxamine solution was added to each sample, which was then incubated in the thermomixer for 90 min at 37 °C. Subsequently, 50 µL of N-Methyl-N-trimethylsilyl trifluoroacetamide (MSTFA) was added, and the samples were incubated for an additional 30 min under the same conditions. A 50 µL aliquot of the final solution was used for the analysis of metabolomic components.

For the preparation of the standard solution and methoxamine solution, ribitol was utilized to prepare the standard solution by dissolving 0.2 mg of ribitol in 1 mL of ultra-pure water. The methoxamine solution was prepared by dissolving 20 mg of methoxamine in 1 mL of pyridine. Metabolomic analysis was performed using gas chromatography–mass spectrometry (GC-MS) with a SHIMADZU system. The GC-MS system comprised the GC-2010 and GCMS-QP2010 Plus (SHIMADZU), and the column used was a DB column (0.25 mm internal diameter, 30 m length, and 1.00 µm film thickness, Agilent Technologies Inc.). The GC conditions were as follows: the inlet temperature was set at 280 °C, and the injection method was split (10:1). The oven temperature was initially held at 60 °C for 1 min, then increased to 320 °C at a rate of 4 °C/min, and held at 320 °C for 10 min, with a helium (He) flow rate of 1.1 mL/min. The MS conditions were as follows: the scanning mode was used for analysis; the transfer line was set at 290 °C; and the ion source was maintained at 200 °C. Mass spectra were recorded at a scan rate of 1 scan/s with a mass-to-charge ratio (*m*/*z*) range of 45–600. Data analysis was conducted using the multivariate analysis software Pirouette version 5.0 (Infometrix, Inc., Bothell, WA, USA).

### 2.12. Statistical Analysis

The statistical analysis was conducted using R software version 4.1.2. Significant differences between the treatment groups were determined using a one-way analysis of variance (ANOVA), with a significance threshold set at *p* < 0.05. To further investigate the differences among group means, post hoc comparisons were performed using Tukey’s Honestly Significant Difference (HSD) test, ensuring that all pairwise comparisons were statistically valid at the *p* < 0.05 level. Additionally, principal component analysis (PCA) was utilized to analyze the metabolome, providing a comprehensive overview of the key metabolites that contributed to the observed variations.

## 3. Results

### 3.1. Fruit Ethylene and CO_2_ Production

Ethylene production in tomato fruits was induced through dropping. The results underscore the critical role of drop height, with a notable impact observed at 100 cm for both green (Figure 1A) and red (Figure 1B) fruits. Ethylene production commenced within one hour, following the dropping event in green fruits. Significantly, tomatoes dropped from a height of 100 cm exhibited the highest levels of ethylene production, followed by fruits dropped from 50 cm, 30 cm, and 10 cm, respectively. Control fruits, which were not subjected to dropping, displayed the lowest ethylene production levels.

The hierarchy of ethylene production levels, from highest to lowest, was observed in fruits dropped from 100 cm, 50 cm, 30 cm, 10 cm, and 0 cm (control), respectively (see Figure 1B for visual comparison). The decreasing trend in ethylene production as dropping height decreased suggests a dose-dependent response to mechanical stress in red fruits. The observation that red fruits showed a decline in ethylene production starting from the second week post-dropping indicates a regulatory mechanism that modulates ethylene biosynthesis over time. This may involve feedback regulation or the exhaustion of ethylene precursor pools, leading to reduced synthesis rates. These findings highlight the intricate interplay between mechanical stimuli, ethylene production, and fruit ripening processes. Understanding these dynamics is crucial for optimizing post-harvest handling practices and enhancing fruit quality and shelf life. Further research into the molecular mechanisms underlying these responses will provide valuable insights into fruit physiology and stress adaptation strategies.

The effect of fruit dropping on ethylene production in red fruit after 1, 6, 24, and 48 h is shown in Figure 2. The results showed that ethylene production was slightly increased by fruit dropping from 30, 50, and 100 cm after 6, 24, and 48 h.

The ethylene production capacity from ACC which presented as ACO (1-Aminocyclopropane-1-Carboxylic Acid Oxidase) in green fruit at 1, 6, 24, and 48 h after dropping is shown in Figure 3A–F. After 1 h, there was no difference among treatments, which means that the enhancement of ethylene shown in Figure 1 was not triggered by the interaction between ACC dissolved in cytosol and ACC oxidase, which binds to the membrane due to the deterioration of membrane integrity caused by dropping [42]. After 6 h, the ethylene production was higher in fruit dropped from 100 cm. At 24 h, there was a trend of increased ethylene production with increased dropping distance, which was observed only at 10 mM ACC. At 48 h, ethylene production increased after dropping from 100 cm in fruit treated with 1 mM and 10 mM ACC. This time course suggests that the enhancement of ethylene production in green fruits 6 h after dropping might result from the de novo gene expression of ACC oxidase [43].

The ethylene production capacity of ACC in red fruit at 1, 6, 24, and 48 h after dropping is displayed in Figure 3. In the red fruit, there was no difference between treatments, which were not affected by fruit dropping nor by ACC concentration.

Fruit dropping significantly affected respiration (CO_2_) in tomato fruits. In both green and red fruits, CO_2_ increased until 24 h after dropping. The peak point of CO_2_ production was 350.1 µL g^−1^ h^−1^, which was observed one day after fruit dropping; however, in red fruits, the peak point of CO_2_ production was 338.2 µL g^−1^ h^−1^, also one day after dropping. In green fruits, the 100 cm dropping treatment resulted in significantly higher CO_2_ production, compared to the control group and other treatments, until 14 days after treatment, but, in red fruits, from 5 days after treatment to 14 days after treatment, there was no significant difference (Figure 4). Considering this, alteration in CO_2_ production was separately measured starting 1 h after dropping to 48 h after dropping (Figure 5).

### 3.2. Fruit Color (a Value)

Dropping exerted a significant influence on the color development of tomato fruits. For green fruits, notable color changes were observed between 5 and 14 days post-treatment (Figure 6A). In red fruits, significant differences in color values (a* value) were recorded on most days, with the exception of the initial day as well as days 1, 2, 3, 4, 12, and 13, when compared to the control (Figure 6B). The highest color values were recorded in fruits subjected to the 100 cm dropping treatment, followed in descending order by those dropped from 50 cm, 30 cm, 10 cm, and the control treatment, for both green and red tomatoes. The rate of color change was notably more rapid in green fruits compared to red fruits. The color values for green fruits ranged from −10 to 35, while, for red fruits, they ranged from 10 to 32.

The visual effect of fruit dropping on green and red tomatoes harvested at the same time under different conditions across a 14-day period is shown in Figure 7. In Set A, green tomatoes on Day 1 appear intact under all conditions, but by Day 7, they start showing signs of damage and color change due to dropping. By Day 14, the damage is more pronounced, with further color change indicating ongoing deterioration. In Set B, red tomatoes also appear intact on Day 1, show some bruising by Day 7, and exhibit more damage and potential shriveling by Day 14. The varying conditions represent different dropping heights or forces, simulating different levels of physical impact. The results suggest that green tomatoes experience immediate and severe damage from dropping, while red tomatoes, although initially more resilient, also deteriorate over time, highlighting how physical impacts affect the quality and shelf life of tomatoes at different ripening stages.

### 3.3. Fruit Hardness

The results indicated that the firmness of tomato fruits stored at 25 °C differed significantly among the treatments. After 14 days of storage, there was a marked decrease in firmness. This decline is attributed to the activity of softening enzymes, which lead to the degradation of cell walls and polysaccharides in the fruit. The treatment involving damage had a notable impact on fruit softening, likely due to an increase in the metabolic rate of the fruit, resulting in water loss. This water loss contributed to the cells becoming flaccid and collapsing under pressure (Figure 8).

### 3.4. Fruit Electrolyte Leakage

Fruit dropping significantly increased electrolyte leakage compared to non-dropped fruits. As illustrated in Figure 9, electrolyte leakage in green fruits exhibited a substantial increase starting 24 h post-dropping, whereas, in red fruits, the increase became significant as early as 6 h after the dropping treatment. The data indicate that, 14 days following the dropping treatments, a height of 100 cm was identified as the critical level, causing severe membrane damage in both green and red fruits. In green fruits, electrolyte leakage increased progressively with higher dropping heights. Conversely, in red fruits, the control treatment resulted in lower electrolyte leakage, with no significant differences observed among the various dropping heights.

### 3.5. Fruit Brix

The impact of fruit dropping on the total soluble solids (TSS) of green tomatoes is illustrated in Figure 10A, while the effect on red tomatoes is depicted in Figure 10B. The results reveal no significant differences among the treatments for red tomato fruits. However, a significant difference was observed in green tomatoes on the first day following the dropping treatment. The Brix level in red fruits, which was approximately 6.57, indicated full ripeness, whereas green fruits had a Brix level of about 5.23. These findings suggest that the stage of maturity at harvest significantly influences the TSS of the fruit.

### 3.6. Fruit Vitamin C

The impact of fruit dropping on vitamin C levels in green tomatoes is illustrated in Figure 11A, while the effect on red tomatoes is depicted in Figure 11B. In green tomatoes, fruit dropping led to a significant decrease in vitamin C levels on the first day post-treatment. However, by the seventh day, dropping significantly (*p* < 0.001) increased vitamin C levels. By the fourteenth day, no significant difference in vitamin C content was observed between the dropped fruits and the control. In red tomatoes, fruit dropping significantly (*p* < 0.001) increased vitamin C levels on days 1 and 7 post-treatment, but these levels decreased by day 14 compared to the control.

### 3.7. PCA of Fruit Metabolome

This PCA biplot illustrates the impact of mechanical stress (dropping) on the metabolite profiles of green and red tomato fruits one week after treatments, with the principal components explaining 82.0% and 64.8% of the variance, respectively.

Green and red tomato samples, subjected to various dropping heights (10 cm, 30 cm, 50 cm, and 100 cm) and controls, are distinctly separated, indicating significant changes in metabolite composition due to the treatments. Control samples for both green and red tomatoes cluster separately, highlighting their unique baseline profiles. Dropped green tomatoes display varied profiles, especially at higher dropping heights (100 cm), showing a pronounced effect. Dropped red tomatoes also show changes, though their central positioning suggests potentially less dramatic shifts compared to green tomatoes. Comparing the dropping effect on metabolites alteration between green and red tomatoes, methionine and ACC (both of which are ethylene precursors) increased in red fruits, corresponding to the elevated release of ethylene. In addition, less aminobutyric acid and more citric acid were found in dropped red fruits. This analysis underscores how dropping treatments significantly alter the biochemical landscape of tomatoes, with certain metabolites being key indicators of stress response (Figure 12). Furthermore, PCA revealed that ACC and methionine levels increased due to fruit dropping. This increase led to higher ethylene production in dropped fruits compared to those that were not dropped (Figure 1 and Figure 2).

## 4. Discussion

### 4.1. Ethylene and CO_2_ Production

Fruit dropping significantly increased ethylene production in both green and red fruits. Dropping from a height of 100 cm was particularly impactful compared to the other treatments. Ethylene production rose rapidly within two days after dropping, so, in a separate experiment, measurements were taken every 6 h for a total of 48 h. The results indicated that, in the first 6 h after dropping, ethylene production increases rapidly; however, this continues up to 48 h and green dropped fruits produced more ethylene. Moreover, principal component analysis revealed that methionine and ACC were induced by fruit dropping. Additionally, higher activity of the ACO enzyme (Figure 3) was observed under fruit dropping treatments. This increased activity might lead to higher ethylene production, as ACO converts ACC to ethylene [44]. This means that having higher ACO activity can convert more ACC to ethylene. Ethylene production as an effect of mechanical stress in tomato fruits was first reported by Macleod et al. [45]. Furthermore, Moretti et al. [14] observed a transient increase in ethylene and CO_2_ production by tomatoes subjected to impacts. However, Poliana et al. [46] indicated that there was no significant change in CO_2_ production in compressed fruits compared to normal ones. In this study, CO_2_ was affected by fruit dropping and significantly induced in stressed fruits compared to control (Figure 4 and Figure 5). The induced ethylene production in dropped fruits might exert positive feedback effects, which regulate the activity of ACC synthase (ACS) enzyme that can also accelerate ethylene production [47].

### 4.2. Fruit Color

A significant change in the redness of tomato fruits was observed, depending on different heights from which the fruits were dropped. A similar result was reported by Al-Dairi et al. [6], who indicated that transportation had a significant effect on the redness of tomato fruits stored at 22 °C; red color, expressed as a* value, changed from 20.05 to 32.21 between the initial day and day 12, respectively. In the present study, the fruits were stored at 25 °C after dropping and the red color change ranged from −10.35 to 30.79 in green fruits and from 11.64 to 31.83 in red fruits. Pathare et al. [48] also reported that the factors such as storage temperature, storage duration, and vibration significantly increased redness in tomato fruits, with the color changing from 38.66% (no vibration) to 54.14% (vibration treatment) after 10 days of storage at 22 °C. Also, Dagdelen and Aday [49] reported a significant a* value color change in peach fruits as an effect of vibration. Furthermore, it has been indicated that high temperature also accelerates fruit reddening during storage [50]. In the present study, 25 °C was chosen because it is the average room temperature without any refrigeration or machinery. The results also revealed a significant positive (*p* < 0.001) relationship between ethylene production and fruit pigmentation. Fruits with more redness produced more ethylene compared to those with less redness. Gulab et al. [51] also reported that ethylene production increased with color development.

### 4.3. Fruit Firmness

The results shown in Figure 8 indicate that fruit dropping has a severe effect on tomato fruit firmness. Both the green and red fruits start to lose their firmness right after dropping. The red fruits were 75.9% softer on day 14 compared to the initial day, while the green fruits were 64.8% softer on day 14 compared to the initial day. This softness might be due to an increase in the activity of the enzyme 1-aminocyclopropane-1-carboxylic acid oxidase (ACO) (Figure 3), which leads to higher ethylene production; the increase in ethylene then triggers softening enzymes. Pathare and Al-Dairi [48] also reported reduced firmness in tomato fruits subjected to vibration, and stated that this might be attributed to enzyme activity [46]. Similar trends were observed by Cherono and Workneh [10], Kabir et al. [52], and Paiva et al. [53]. In the present study, a 100 cm dropping height significantly decreased fruit firmness, and, in the meantime, ACO activity was two times higher in green than in red fruits.

### 4.4. Electrolyte Leakage

Fruit dropping significantly induced (*p* < 0.001) electrolyte leakage in both green and red tomato fruits. The highest electrolyte leakage was observed 48 h after dropping (Figure 9A,C). Moreover, the results showed that, 14 days after dropping, all dropping treatments produced higher electrolyte leakage than the control (Figure 9B,D). Basically, stress damages fruits’ cell walls and, as a result, electrons leak from the cells. K^+^ electrons are a main constituent, which may induce cell death [54]. Pennazio and Sapetti [55] reported that mechanical injuries increased electrolyte leakage in cowpea leaves, which suggests that the permeability alterations occurred in advance of cell death, and indicated that this alteration is considered to be the primary response to stress.

### 4.5. Brix

Brix levels significantly increased (*p* < 0.001) with storage time in red fruits; however, after 14 days, there was no difference in the Brix content of green fruits compared to the initial day. This indicates that, in red fruits, pectin or other polysaccharides are changed into simple sugars; however, the sugar content in green fruits is low. The increase in Brix levels in the green fruits one day after treatment might be due to water loss [56,57,58]. In green fruits, Brix levels ranged from 4.5 to 6.2, while, in red fruits, they ranged from 5.1 to 7.2. Montero et al. [59] also reported increases in Brix levels as an effect of mechanical stress in red tomato fruits. However, in Mandarin, it has been reported that, with storage time, Brix levels are reduced, due to mechanical stress [60]. Moreover, it has been observed that mechanical stress induces the activity of enzymes such as polygalacturonase and pectin methylesterase, accelerating the breakdown of sugars and fruit softening [60,61]. This causes a decrease in fruit firmness (Figure 8) and induces increased fruit Brix levels (Figure 10), a result that the current study also confirms.

### 4.6. Vitamin C

Vitamin C is an important multifaceted phytonutrient and radical scavenger which is required for plant growth; a suitable amount of it is needed for a normal postharvest plant physiology [62,63]. Vitamin C in green tomato fruits subjected to dropping decreased (*p* < 0.001) compared to the control; on day 7, it increased (*p* < 0.001), but, on day 14, the difference was not significant. In red fruits, mechanical stress induced a vitamin C increase on day 1 and day 7 after dropping, but, on day 14, vitamin C was decreased (*p* < 0.001) in dropping treatments compared to the control (Figure 11). Ntagkas et al. [64] observed that the amount of vitamin C was higher in fruits with a higher amount of carbohydrates, which is similar to the current study’s result. Sablani et al. [65] reported that vitamin C increased in tomato fruits with storage time but no significant difference was observed between damaged and undamaged fruits. In leafy vegetables, mechanical stress had a positive effect on vitamin C [66], due to the low amount of sugars compared to fruits.

### 4.7. Metabolomic Analysis

Metabolomic analysis revealed that fruit dropping affected the metabolites of both green and red tomatoes. Dropped fruits had higher sugar levels and lower acid levels compared to control fruits. Furthermore, methionine and ACC (Figure 12) were higher in dropped fruits, leading to increased ethylene production and more damage compared to the controls (Figure 12). Wang et al. [67] reported that dropped citrus fruits showed a higher number of phytochemicals in a metabolomic analysis. In the present study, metabolomic analysis showed that fruit dropping caused an increase in methionine and ACC, which are the precursors of ethylene, a hormone that regulates ripening. Increased levels of methionine and ACC can lead to higher ethylene production, which may accelerate the ripening process and affect the timing of fruit softening and color changes. Yin et al. [68] reported metabolic alteration as follows: aminobutyric acid →→→ succinate → fumaric acid → malic acid → oxalic acid → phosphoenolpyruvic acid → pyruvic acid →→ citric acid, following the course of fruit maturation. Therefore, the decreased aminobutyric acid levels and increased citric acid levels in dropped fruits observed in our study indicate the promotion of maturation processes. The increase in galacturonic acid in both green and red dropped fruits might result from the enhanced activity of polygalacturonase, a softening enzyme. There are few studies on the effects of fruit dropping on metabolites, and no previous research has specifically investigated the impact on tomato fruit metabolites.

## 5. Conclusions

The results demonstrate that mechanical stress from fruit dropping significantly impacts ethylene production and various physiological and biochemical attributes of tomatoes. Dropping from 100 cm induced the highest ethylene levels in both green and red tomatoes, with a rapid increase in ethylene and CO_2_ production within hours, especially in green fruits. This stress also affected fruit color, firmness, and electrolyte leakage, with the most severe damage observed in fruits dropped from 100 cm. Brix levels rose in red fruits but remained stable in green tomatoes, while vitamin C content fluctuated but stabilized over time after the dropping treatments. Metabolomic analysis revealed notable changes in metabolite profiles, with increased sugars and amino acids like methionine and ACC linked to stress responses. Principal component analysis confirmed significant metabolic alterations in dropped fruits. These findings highlight the need to minimize physical impacts during post-harvest handling to maintain fruit quality and shelf life and suggest that further research into the underlying mechanisms could improve post-harvest management practices.

## Figures and Tables

**Figure 1 biomolecules-14-01012-f001:**
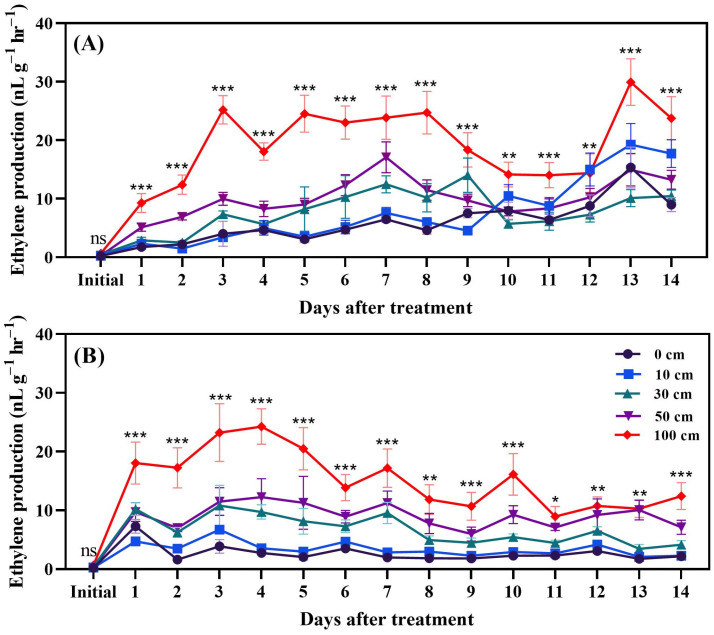
Ethylene production of green (**A**) and red fruits (**B**) after dropping from different heights. The significance level was determined to be *p* < 0.05, with *, **, and *** indicating *p* < 0.05, *p* < 0.01, and *p* < 0.001, respectively. “ns” was used to indicate no significance.

**Figure 2 biomolecules-14-01012-f002:**
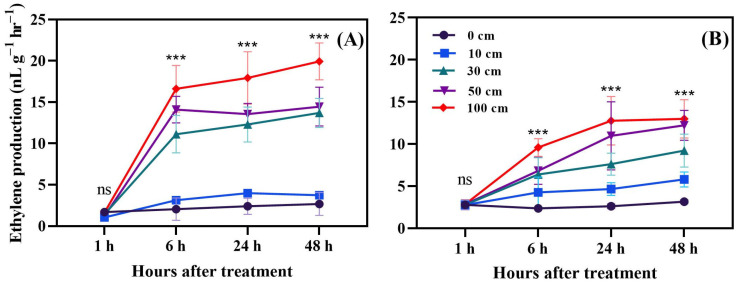
Hourly ethylene production measurements from green (**A**) and red (**B**) tomato fruits after being subjected to dropping treatments. The significance level was determined to be *p* < 0.05, with *** indicating *p* < 0.001. “ns” was used to indicate no significance.

**Figure 3 biomolecules-14-01012-f003:**
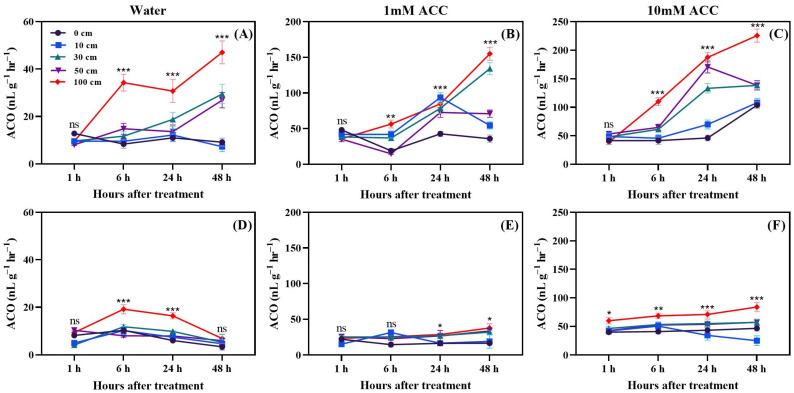
Effects of dropping on ACO production in green tomato fruits ((**A**) in water, (**B**) in 1 mM ACC solution, and (**C**) in 10 mM ACC solution) and red tomato fruits ((**D**) in water, (**E**) in 1 mM ACC solution, and (**F**) in 10 mM ACC solution). The significance level was determined to be *p* < 0.05, with *, **, and *** indicating *p* < 0.05, *p* < 0.01, and *p* < 0.001, respectively. “ns” was used to indicate no significance.

**Figure 4 biomolecules-14-01012-f004:**
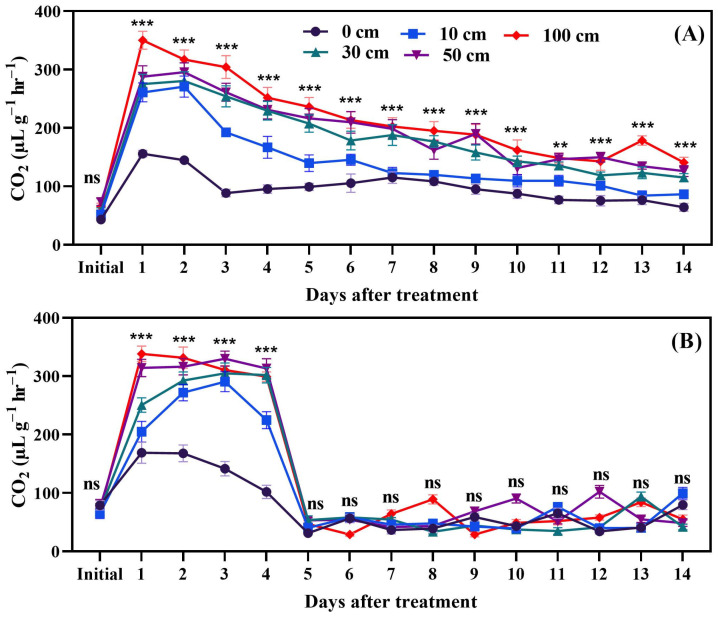
CO_2_ production of green (**A**) and red fruits (**B**) after dropping from different heights. The significance level was determined to be *p* < 0.05, with **, and *** indicating *p* < 0.01, and *p* < 0.001, respectively. “ns” was used to indicate no significance.

**Figure 5 biomolecules-14-01012-f005:**
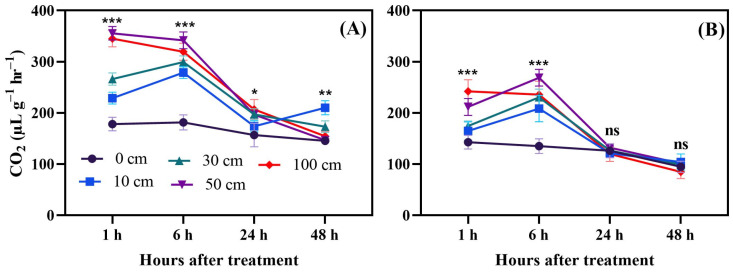
Hourly CO_2_ production measurements from green (**A**) and red (**B**) tomato fruits after being subjected to dropping treatments. The significance level was determined to be *p* < 0.05, with *, **, and *** indicating *p* < 0.05, *p* < 0.01, and *p* < 0.001, respectively. “ns” was used to indicate no significance.

**Figure 6 biomolecules-14-01012-f006:**
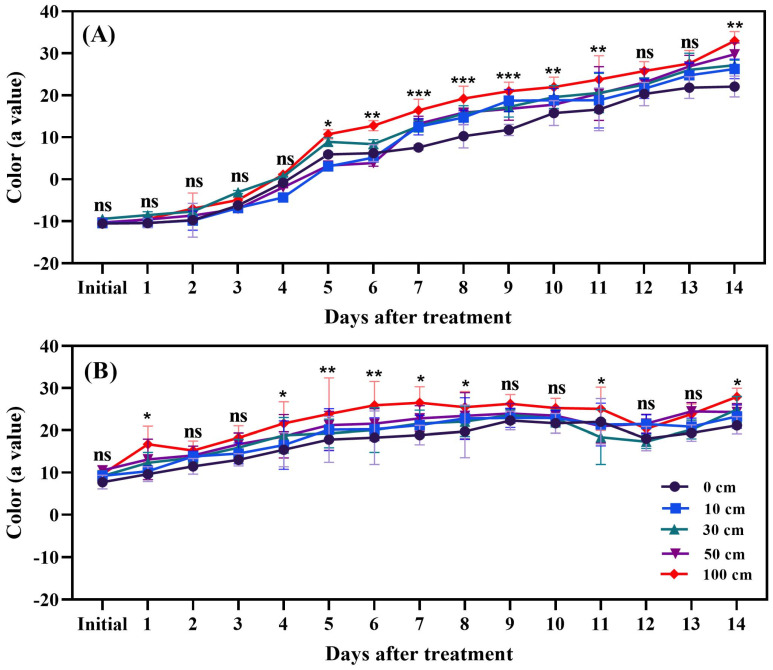
The color changes in green (**A**) and red (**B**) tomato fruits after the application of dropping treatments. The significance level was determined to be *p* < 0.05, with *, **, and *** indicating *p* < 0.05, *p* < 0.01, and *p* < 0.001, respectively. “ns” was used to indicate no significance.

**Figure 7 biomolecules-14-01012-f007:**
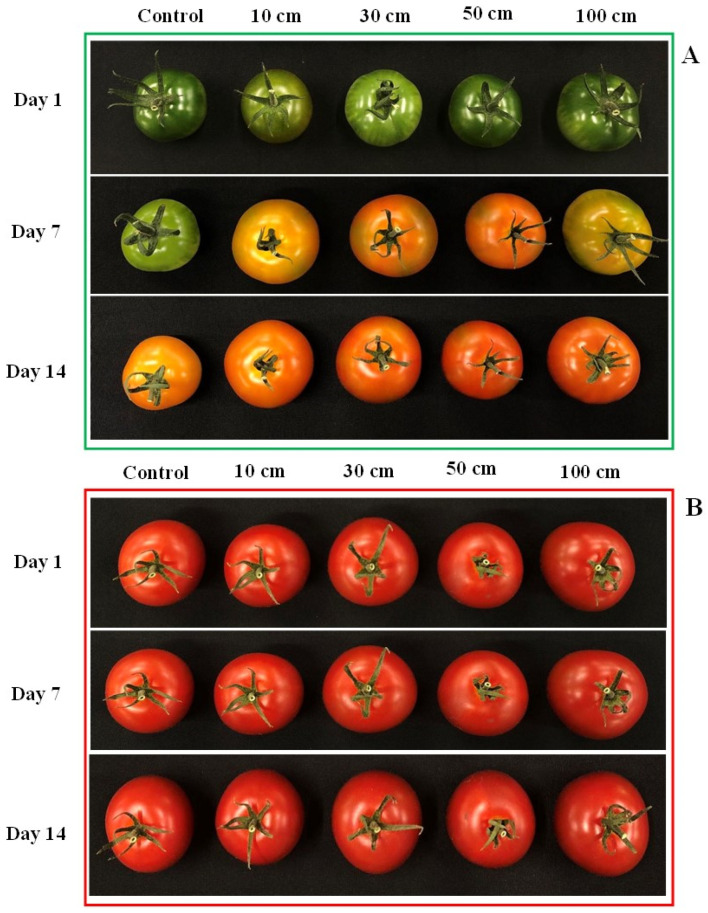
Visual impact of dropping treatments on green (**A**) and red (**B**) tomato fruits.

**Figure 8 biomolecules-14-01012-f008:**
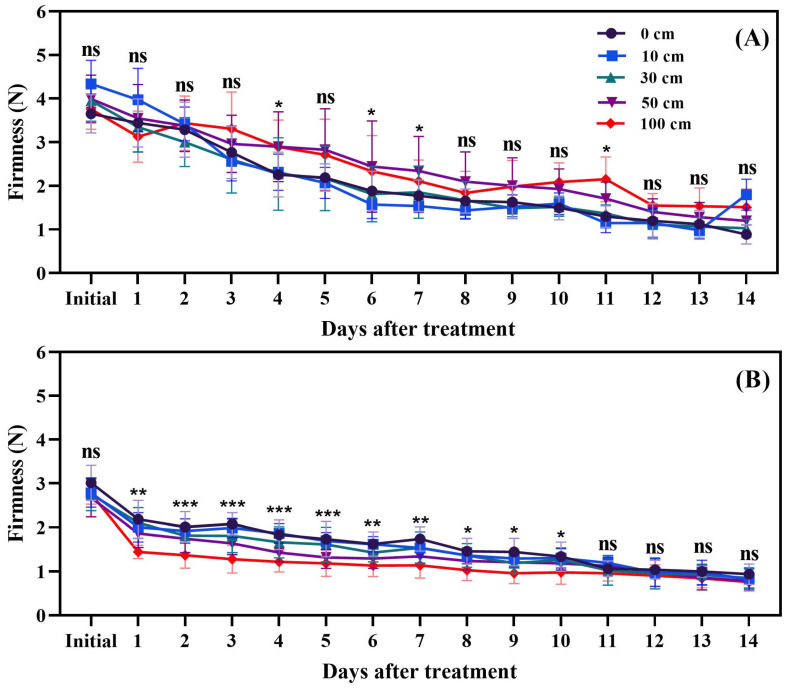
Impact of dropping on the firmness of green (**A**) and red (**B**) tomato fruits. The significance level was determined to be *p* < 0.05, with *, **, and *** indicating *p* < 0.05, *p* < 0.01, and *p* < 0.001, respectively. “ns” was used to indicate no significance.

**Figure 9 biomolecules-14-01012-f009:**
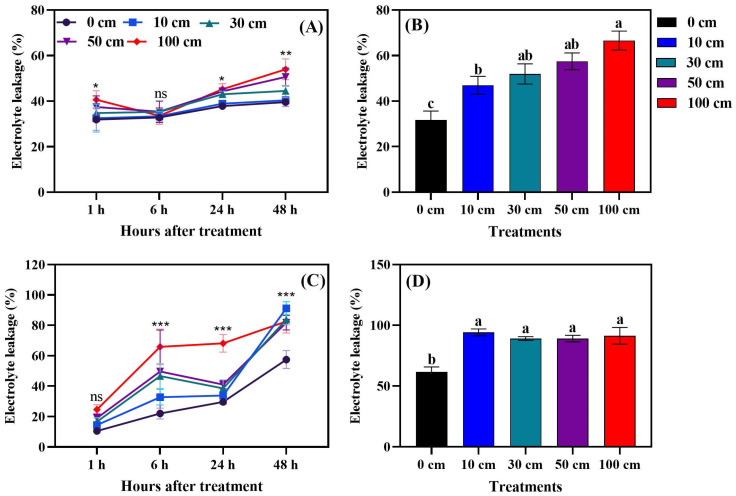
Hourly changes in electrolyte leakage of green (**A**) and red (**C**) tomato fruits, and after 14 days of dropping for green (**B**) and red (**D**) tomato fruits. The significance level was determined to be *p* < 0.05, with *, **, and *** indicating *p* < 0.05, *p* < 0.01, and *p* < 0.001, respectively. “ns” was used to indicate no significance. Different letters above SD indicate Tukey’s test at a significance level of 0.05.

**Figure 10 biomolecules-14-01012-f010:**
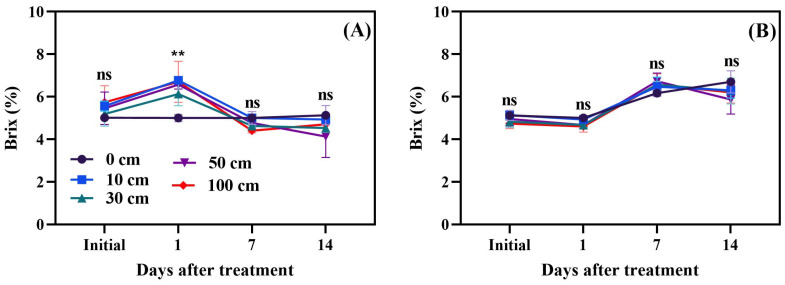
Assessment of Brix levels in green (**A**) and red (**B**) tomato fruits following dropping treatments. The significance level was determined to be *p* < 0.05, with ** indicating *p* < 0.01. “ns” was used to indicate no significance.

**Figure 11 biomolecules-14-01012-f011:**
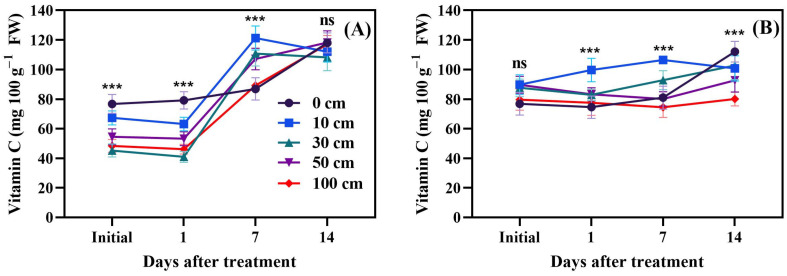
Influence of dropping on the vitamin C content in green (**A**) and red (**B**) tomato fruits. The significance level was determined to be *p* < 0.05, with *** indicating *p* < 0.001. “ns” was used to indicate no significance.

**Figure 12 biomolecules-14-01012-f012:**
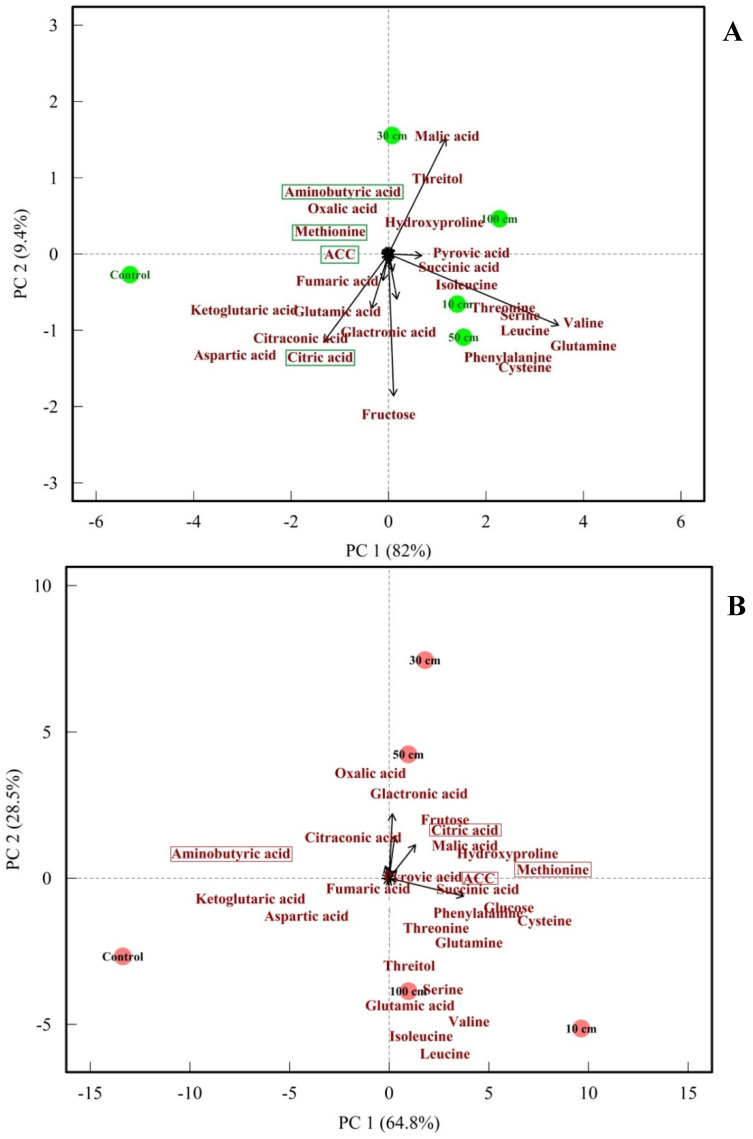
Principal component analysis of metabolites in green and red tomato fruits. The circles represent dropping treatments: green circles for green fruits (**A**) and red circles for red fruits (**B**).

## Data Availability

Data will be available upon request from the corresponding author.

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
