# Peer review of "Impact of Dropping on Postharvest Physiology of Tomato Fruits Harvested at Green and Red Ripeness Stages"

_biomolecules, 2024, doi:10.3390/biom14081012_

Round 1

Reviewer 1 Report

Comments and Suggestions for Authors

Authors report the impact of dropping on postharvest physiology of tomatoes, studying multiple physical, physiological, and quality attributes. The work presented is of interest, given the occurrence of fruit dropping. Below are some comments for authors:

- Some of the figures need to be corrected, on Fig. 9.D) says "Hours after treatment", while it should be "Treatments"

- Some of the experimental setup need further explanation. On "4.7. Fruit firmness measurement", do not show information on the penetrometer used.

- On Figure 10, it is shown that brix (%) are significantly different on the treatment group compared to the control on day 1, before dropping to the control levels afterwards. Can authors expand on the explanation of why this happens.

Reviewer 2 Report

Comments and Suggestions for Authors

This article is based on the post-harvesting impact on the fruit quality owing to dropping. 

The abstract of this work is irrelevant and I did not see any new information.  Only provide the relevant information related to the study. Reduce the section length of Materials and Methods. Provide the uncertainty in the results that happened during the analysis. Also provide the relevant spacing between the symbol and magnitude. The section 4.12 is not sufficient. Kindly provide the proper detail of statistical deviation.  p value ? What about F-test ?

Comments on the Quality of English Language

Need to restructure the sentences. 

Author Response

Please find the attached file containing detailed responses to each of your valuable comments. We have addressed your feedback point-by-point and made the necessary revisions to the manuscript. Thank you for your insightful review.

Round 2

Reviewer 2 Report

Comments and Suggestions for Authors

It can be considered. 

Comments on the Quality of English Language

Fine.